# FOXO3 Expression in Macrophages Is Lowered by a High-Fat Diet and Regulates Colonic Inflammation and Tumorigenesis

**DOI:** 10.3390/metabo12030250

**Published:** 2022-03-16

**Authors:** Rida Iftikhar, Harrison M. Penrose, Angelle N. King, Yunah Kim, Emmanuelle Ruiz, Emad Kandil, Heather L. Machado, Suzana D. Savkovic

**Affiliations:** 1Department of Pathology and Laboratory Medicine, School of Medicine, Tulane University, New Orleans, LA 70012, USA; riftikhar@tulane.edu (R.I.); hpenrose@tulane.edu (H.M.P.); aking22@tulane.edu (A.N.K.); ykim8@tulane.edu (Y.K.); 2Division of Endocrine and Oncologic Surgery, Department of Surgery, Tulane University, New Orleans, LA 70012, USA; remmanuelle@tulane.edu (E.R.); ekandil@tulane.edu (E.K.); 3Department of Biochemistry and Molecular Biology, School of Medicine, Tulane University, New Orleans, LA 70012, USA; hmachado@tulane.edu

**Keywords:** macrophages, FOXO3, obesity, inflammation, tumorigenesis, IBD, colon cancer

## Abstract

Obesity, characterized by augmented inflammation and tumorigenesis, is linked to genetic predispositions, such as FOXO3 polymorphisms. As obesity is associated with aberrant macrophages infiltrating different tissues, including the colon, we aimed to identify FOXO3-dependent transcriptomic changes in macrophages that drive obesity-mediated colonic inflammation and tumorigenesis. We found that in mouse colon, high-fat-diet-(HFD)-related obesity led to diminished FOXO3 levels and increased macrophages. Transcriptomic analysis of mouse peritoneal FOXO3-deficient macrophages showed significant differentially expressed genes (DEGs; FDR < 0.05) similar to HFD obese colons. These DEG-related pathways, linked to mouse colonic inflammation and tumorigenesis, were similar to those in inflammatory bowel disease (IBD) and human colon cancer. Additionally, we identified a specific transcriptional signature for the macrophage-FOXO3 axis (MAC-FOXO3_82_), which separated the transcriptome of affected tissue from control in both IBD (*p =* 5.2 × 10^−^^8^ and colon cancer (*p =* 1.9 × 10^−^^11^), revealing its significance in human colonic pathobiologies. Further, we identified (heatmap) and validated (qPCR) DEGs specific to FOXO3-deficient macrophages with established roles both in IBD and colon cancer (IL-1B, CXCR2, S100A8, S100A9, and TREM1) and those with unexamined roles in these colonic pathobiologies (STRA6, SERPINH1, LAMB1, NFE2L3, OLR1, DNAJC28 and VSIG10). These findings establish an important understanding of how HFD obesity and related metabolites promote colonic pathobiologies.

## 1. Introduction

Obesity is characterized by a systemic inflammatory state and augmented tumorigenesis in various tissues, including the colon [1,2,3]. In the human population, obesity is accompanied by genetic predispositions such as polymorphisms of the *FOXO3* gene that lead to its lowered levels [4]. FOXO3, a transcription factor, plays a central role in diverse cellular functions in colonic and immune cells, including macrophages [5,6]. Therefore, elucidating how obesity, through loss of FOXO3 in macrophages, promotes colonic inflammation and tumorigenesis will provide conceptual advances in understanding this worldwide epidemic and the mechanisms driving these human pathobiologies.

Macrophages are critical for tissue homeostasis and repair [7,8]. Their aberrant function promotes inflammation and tumorigenesis [9,10]. Depending on the stimuli, macrophages polarize towards different subpopulations with distinct characteristics and functions [11,12]. Further, tissue microenvironments, such as those found in the adipose tissue of obese individuals and in colon tissue affected by infection, inflammation, and tumorigenesis, contribute to the polarization of macrophages [13,14]. Additionally, macrophages release cytokines which result in tissue damage that exacerbates chronic inflammation [15]. In inflammatory bowel disease (IBD), aberrant macrophages (enriched for CD68^+^ and CD14^hi^CD16^+^ phenotypes) in the inflamed intestine promote cytokine production and delayed bacterial clearance, thereby promoting tissue injury and worsening disease outcome [7,16,17]. Moreover, macrophages promote tumor progression through angiogenesis, intravasation and metastasis [18,19]. Breast and pancreatic tumor expansion and metastasis are facilitated by extracellular matrix remodeling, mediated by macrophages [20,21]. In colonic tumors, increased macrophage infiltration with M1/NOS2^+^ or M2/CD163^+^ phenotypes is associated with cancer aggressiveness, growth, and poor survival rates [22,23]. Therefore, it is plausible that with obesity mediators and metabolites, aberrant peritoneal macrophages are a critical player in promoting colonic inflammation and tumorigenesis.

Forkhead Box O3 (FOXO3) transcription factor has diverse cellular functions, including growth, immune response, and metabolic homeostasis [24,25,26]. In humans, obesity is associated with polymorphisms in the FOXO3 gene [4]. Further, a high-fat diet (HFD) shortens the lifespan of mice by diminishing FOXO3 levels in the central nervous system [27]. Obesity mediators and metabolites cause a loss of FOXO3 in colonic cells [28,29], revealing an important role of FOXO3 in obesity-mediated changes in the colon. Additionally, in colonic cells, bacterial products and inflammatory mediators abrogate FOXO3 function to further promote inflammation [30,31]. Genome-wide association studies (GWAS) identified a polymorphism in FOXO3 that is associated with IBD severity [32]. FOXO3 also plays a critical role in immune cells, and its deficiency promotes the activation of T cells and macrophages [33,34]. Moreover, FOXO3 acts as a tumor suppressor, and its loss of activity has been shown to be closely associated with cancer initiation and progression [35]. In human colon cancer, markedly reduced FOXO3 levels correlate to advanced tumors [36]. We have previously shown in mice that FOXO3 deficiency promotes the development and progression of inflammatory colon cancer [28,30,31,37,38]. Additionally, the colons of FOXO3-deficient mice have an enrichment in macrophages and elevated levels of intracellular lipid droplets [29,38]. Therefore, it is plausible that the loss of FOXO3 in both colonic cells and macrophages may be critical in driving obesity-mediated colonic inflammation and tumorigenesis.

Here, we established the significance of the loss of FOXO3-dependent functions of macrophages in intestinal inflammation and tumorigenesis. These findings will provide conceptual advances in understanding how obesity (mediators and metabolites) promotes these colonic pathobiologies.

## 2. Results

### 2.1. HFD Obesity in Mice Mediates Increased Presence of Macrophages and Loss of FOXO3 in Colon

Obesity is associated with aberrant macrophages present in different tissues, which may drive colonic pathobiologies [39,40]. In human populations, obesity is associated with polymorphisms in the FOXO3 gene [4]. Additionally, obesity mediators and metabolites cause a loss of FOXO3 in colonic cells [28,29]. Therefore, we hypothesized that the loss of FOXO3 via obesity contributes to the development of pathobiologies in various tissues, especially the colon. First, we utilized transcriptomic data obtained from the colon of both HFD obese mice and mice fed with regular diet (RD) (SRP093363). As we described before [41], consistent with an obese phenotype, mice on a HFD gained, on average, 54 ± 2 * g, while mice on a RD gained 33 ± 3 g during a period of 20 weeks (* *p <* 0.05). From these mice, we utilized transcriptomes obtained from their colon [41] to assess macrophage levels with CIBERSORT [42]. This is a platform that employs the transcriptional signatures of specific immune cells to determine their levels in the tissue. There was an abundance of macrophages in the colon of HFD obese mice relative to control (RD) (Figure 1A). Moreover, we assessed scraped mucosa from the colon of HFD obese and RD control mice for FOXO3 status. We found increased phosphorylated (inactive) FOXO3 levels in HFD obese mice colonic mucosa relative to control (Figure 1B). Therefore, as HFD obesity causes increased macrophages as well as loss of FOXO3 in mouse colon, we determined that the loss of FOXO3 in macrophages contributes to obesity-mediated colonic pathobiology. We obtained intraperitoneal macrophages [43,44] from FOXO3 knockout (KO) and wildtype (WT) mice in order to determine systemic transcriptional changes (RNA-seq). The population of these induced peritoneal macrophages included a slight number of other immune cells [43,45] which, after further sequencing and bioinformatic analysis, proved to be insignificant. Transcriptional assessment showed a significant number of differentially expressed genes (DEGs) in macrophages obtained from FOXO3-deficient mice relative to control (WT). Among these DEGs, 501 were increased, and 380 were decreased (Figure 1C) (>|1.5|-fold change, FDR < 0.05). The diseases and functions associated with these DEGs include cancer, gastrointestinal diseases, inflammatory response, and lipid metabolism (Figure 1D). Further, pathways associated with these DEGs were compared to transcriptomes obtained from the colon of HFD obese mice. We found similarities in pathways and upstream regulators of FOXO3-deficient macrophages with HFD obese mice colon associated with inflammation (IL-1A, IL-1B, IL-6, IL-8 signaling), growth (ILK signaling, CDK5 signaling, VEGF signaling) and cancer (cAMP-mediated signaling) (Figure 1E,F). Together, these data revealed that systemic transcriptional changes, mediated by the loss of FOXO3 in macrophages, are similar to those seen in the colon of HFD obese mice, which suggests a critical role of the macrophage-FOXO3 axis in driving obesity-mediated colonic changes.

### 2.2. FOXO3 Deficiency in Macrophages Is Associated with Colonic Inflammation and Cancer

To determine whether FOXO3 deficiency in macrophages contributes to colonic inflammatory and tumorigenic processes, we assessed publicly available transcriptomes obtained from mouse colon with inflammation and dysplasia. Pathways and upstream regulators representing transcriptomes of FOXO3-deficient macrophages showed strong similarity to those related to inflammation and tumorigenesis in mouse colon (GSE31106) (Figure 2A–D). Next, we assessed publicly available transcriptomes from IBD and colon cancer patient cohorts to determine the significance of FOXO3-deficient macrophages in human colonic inflammation and tumorigenesis. We found that transcriptomes of FOXO3-deficient macrophages represented similar alterations in pathways and upstream regulators as those seen in IBD (GSE4183) and colon cancer (GSE4183, GSE141174). Specifically, these shared pathways with IBD were linked to inflammation (TNF, IL-1A, IL-1B, NFkB, IL-6) and growth (TREM1, TGFB1) (Figure 3A,B), while other pathways shared with colon cancer were associated with colorectal metastatic signaling, growth (TREM1, NFkB and HMGB1) and inflammation (IL-8, IL-6) (Figure 3C,D). These data demonstrated that FOXO3 deficiency in macrophages is associated with colonic inflammatory and tumorigenic pathobiologies.

### 2.3. FOXO3-Deficient Macrophage Signature Is Prevalent in Human Colonic Inflammation and Cancer

Next, to determine the significance of the macrophage-FOXO3 axis in colonic pathobiologies, we established a transcriptional signature for FOXO3-deficient peritoneal macrophages. A transcriptional panel of DEGs from FOXO3-deficient macrophages relative to control (WT) was generated by calculating fold change between the 2 groups, meeting stringent differential expression and statistical thresholds of log_2_ fold-change > |1.5| and an adjusted *p*-value < 0.001. The panel was comprised of 82 DEGs (MAC-FOXO3_82_), among which 65 were upregulated and 17 were downregulated in FOXO3-deficient macrophages relative to control (Table 1). Principal component analysis (PCA) showed that MAC-FOXO3_82_ separated the transcriptomes from IBD samples according to their inflamed and control (non-inflamed) states with high significance (*p =* 5.2 × 10^−8^) (GSE4183, Figure 4A). Further, unsupervised hierarchal clustering of MAC-FOXO3_82_ separated transcriptomes from IBD samples into inflamed and non-inflamed groups (Figure 4B). Additionally, MAC-FOXO3_82_ separated cancer samples from normal with a high significance, as shown by PCA (*p =* 1.9 × 10^−11^) and unsupervised hierarchal clustering (TCGA, Figure 4C,D). In colon cancer patients, Kaplan–Meier estimates showed that increased MAC-FOXO3_82_ presence is associated with poor survival rates, increased risk of cancer recurrence, and distant metastasis (TCGA, Figure 4E). Together, these data demonstrated that the macrophage-FOXO3 axis is associated with both human IBD and colon cancer.

### 2.4. Expression of Select FOXO3-Dependent Genes in Peritoneal Macrophages

Next, we identified the top DEGs specific to FOXO3-deficient peritoneal macrophages relative to control shown in a heatmap, Figure 5A, many of which are also represented in MAC-FOXO3_82_. These DEGs were validated in FOXO3-deficient macrophages (vs WT) by qPCR, and their status was determined in publicly available transcriptomes of IBD and human colon cancer cohorts. It is important to consider that these DEGs may vary in expression in different subtypes of macrophages or in other cell types. We found these DEGs with established roles in human colonic inflammation and tumorigenesis such as Interleukin 1 Beta (IL-1B), C-X-C Motif Chemokine Receptor 2 (CXCR2), S100 Calcium-binding Protein A8 (S100A8), S100 Calcium-binding Protein A9 (S100A9), and Triggering Receptor Expressed On Myeloid Cells 1 (TREM1) to be significantly increased in FOXO3-deficient macrophages (Figure 5B). Similar alterations in these DEGs were also seen in publicly available transcriptomes from human IBD and colon cancer (Figure 5C,D). These findings reveal that FOXO3 deficiency in macrophages significantly contributes to systemic transcriptional alteration found in IBD and colon cancer.

Furthermore, we validated DEGs in FOXO3-deficient macrophages, whose role in IBD and colon cancer is not yet well understood, including Stimulated By Retinoic Acid Gene 6 Protein Homolog (STRA6), Serpin Family H Member 1 (SERPINH1), Laminin Subunit Beta 1 (LAMB1), Oxidized Low-Density Lipoprotein Receptor 1 (OLR1), Nuclear Factor, Erythroid 2-like 3 (NFE2L3), DnaJ Heat Shock Protein Family (Hsp40) Member C28 (DNAJC28) and V-Set And Immunoglobulin Domain-containing 10 (VSIG10) (Figure 6A). These novel DEGs showed similar alterations in transcriptomes of human IBD and colon cancer, suggesting that they may play a role in these human pathobiologies (Figure 6B,C). These data establish loss of FOXO3-dependent novel genes in macrophages that may regulate these human pathobiologies.

## 3. Discussion

Obesity is a chronic inflammatory state associated with increased risk and progression of colon cancer [1,2,3]. Here, we demonstrated that the loss of FOXO3 in macrophages plays a critical role in obesity-mediated inflammation and tumorigenesis in the colon. We identified that the transcriptome of FOXO3-deficient peritoneal macrophages, similar to the transcriptome from mouse HFD obese colon, shared pathways with colonic inflammation and tumorigenesis. Furthermore, we showed the significance of a MAC-FOXO3_82_ transcriptional signature in human colonic inflammation (seen in IBD) and tumorigenesis (in colon cancer). Ultimately, we identified differentially expressed genes from FOXO3-deficient macrophages with established roles in both IBD and colon cancer, as well as novel genes whose roles in these colonic pathobiologies are not well understood. Together, these findings establish the significance of the loss of FOXO3 in macrophages in colonic pathobiologies and provide conceptual advances in our understanding of how obesity, via both its mediators and metabolites, promotes these disease processes.

We identified, in macrophages, alterations in gene expression dependent on the loss of FOXO3 that are associated with colonic inflammation and tumorigenesis. Macrophages play a central role in tissue homeostasis, including the colon [8,46]. Their polarization is accompanied by metabolic reprogramming, such as the pro-inflammatory subtype mainly shifting to glycolysis, whereas the anti-inflammatory subtype shifts to mitochondrial oxidative phosphorylation [47]. Furthermore, altered lipid metabolism in macrophages is associated with their inflammatory response [47], supporting their direct contribution to the immune-metabolic axis within tissues. Indirectly, these macrophages, by releasing cytokines and metabolites, can further facilitate colonic cells’ inflammatory and tumorigenic responses [8,46,48]. Increased malate, fumarate, citrate and glutamate metabolites in macrophages [47] can augment the immuno-metabolic axis in tissue. Moreover, as FOXO3 emerges as an important regulator of macrophage function [6,49] and controls metabolism in various cells [25,26], it is plausible that FOXO3 plays a central role in metabolic reprograming in macrophages. This critical immuno-metabolic function of FOXO3 is also seen in colonic cells [30,31]. Further, altered macrophages can impair barrier function, as seen in IBD, in addition to promoting the tumor microenvironment [16,50]. Our study showed that obesity mediators and metabolites lead to diminished levels of FOXO3 and increased macrophages in the colon, suggesting a central role of FOXO3 in metabolic reprogramming of both macrophages and intestinal epithelial cells. Consequently, macrophage metabolism is often abnormal in disease states, and their loss of FOXO3-mediated metabolic reprogramming presents a potential therapeutic target for colonic inflammation and tumorigenesis facilitated by obesity.

We validated select differentially expressed genes of FOXO3-deficient macrophages with established roles both in IBD and colon cancer. Increased IL-1B, a potent inflammatory mediator, is associated with promoting colonic inflammation and tumorigenesis [51,52]. In human cartilage, IL-1B causes a loss of FOXO3 [53], which suggests feedback between IL-1B and FOXO3. Further, CXCR2, essential for the maintenance, survival, and self-renewal of hematopoietic cells [54], is elevated with FOXO3 deficiency in macrophages. CXCR2 contributes to inflammation and tumorigenesis in several tissues, including the colon [55,56]. Additionally, S100A8/A9 plays a significant role in colonic inflammatory response and tumorigenesis [57,58]. Their high levels in inflamed tissue position them as a potential biomarker for IBD [59]. Further, TREM1, a membrane receptor in macrophages, drives the innate immune response against bacterial products, thus worsening the course of IBD [60]. In lung and colon cancer, increased TREM1 promotes tumorigenesis [61,62]. These findings establish a critical role of FOXO3 in macrophages, driving colonic inflammation and cancer.

We also identified several differentially expressed genes in FOXO3-deficient macrophages with unexamined roles in colonic inflammation and tumorigenesis. STRA6 regulates the homeostasis and uptake of retinol [63]. HFD-induced STRA6 expression in adipose tissue increases the secretion of inflammatory mediators [64]. As retinoic acid is important for immune cells associated with IBD [65], STRA6 is a novel regulator in these inflammatory processes. Further, polymorphisms in the STRA6 gene are associated with increased incidences and aggressive courses of non-small cell lung cancer [66]. One study shows that in mouse colon, HFD-mediated increased STRA6 may promote tumorigenesis via cancer stem cell populations [67]. Moreover, SERPINH1 is important in the regulation of the extracellular matrix [68]. Recent studies showed that in mouse models of colonic inflammation, SERPINH1 is expressed in infiltrating immune cells and plays a role in extracellular matrix remodeling that promotes colon cancer progression [69]. Additionally, LAMB1 is a member of laminins important for the integrity of the epithelial-stromal network. A genome-wide study (GWAS) identified LAMB1 as a possible driver in IBD pathogenesis through impaired barrier function [70,71,72]. One study showed increased blood levels of LAMB1 in colon cancer patients relative to controls, highlighting its potential biomarker properties [73]. Another FOXO3-dependent gene in macrophages, the transcription factor NFE2L3 [74], is implicated in colonic inflammation and tumorigenesis. Increased expression of NFE2L3 is seen in ulcerative colitis and colon cancer [75,76,77]. Furthermore, OLR1, a low-density lipoprotein receptor with immune response function, is found to be expressed in human intestinal epithelial cells and is involved in the regulation of barrier function [78,79]. Increased expression of OLR1 in pancreatic and colon cancer has been associated with metastasis and poor prognosis [80,81,82]. Furthermore, the roles of VSIG10 and DNAJC28, which may be involved in stem cell function [83,84], in IBD and colon cancer, remain unknown. These novel FOXO3-dependent genes in macrophages provide a basis for conceptual advances in diagnostic medicine and targeted therapy in human pathobiologies.

Obesity, associated with systemic inflammation and increased risk and progression of colon cancer, is increasingly prevalent both in the industrialized and developing world [1,2,3]. Obesity mediators and metabolites affect immune cell populations such as macrophages, thereby promoting inflammation and providing a niche for tumor progression [18,19,39,85]. Additionally, obesity has a wider implication in metabolic disorders such as type II diabetes, hypertension and coronary artery diseases in which macrophages play a significant role in the development of inflammatory response and insulin resistance [39,40,86]. Here we showed that obesity, whether directly or indirectly, drives colonic inflammation and tumorigenesis through the inactivation of FOXO3 in macrophages. Additionally, we established the significance of the macrophage-FOXO3 axis in both IBD and colon cancer. In future studies, the novel differentially expressed genes that we have identified, which are FOXO3-dependent in macrophages, will be examined for their roles in these colonic pathobiologies. Moreover, as macrophage subpopulations vary in different tissue, it is important to note that this macrophage-FOXO3 axis can be extrapolated to other macrophage subtypes in obesity-related disorders. Further findings would expand our understanding of mechanisms involved in obesity-related disorders mediated by the macrophage-FOXO3 axis, help us identify key regulators of other metabolic disorders and develop individualized treatment options.

## 4. Materials and Methods

### 4.1. Human IBD and Colon Cancer Samples

Publicly available transcriptomes obtained from inflammatory bowel disease included control and inflamed colon samples (*n* = 23; GSE4183). Additionally, publicly available transcriptomic data was obtained from colon cancer patients from two cohorts, including normal colon and tumors (*n* = 29, GSE4183, GSE141174). Moreover, publicly available transcriptomic data of colon cancer patients, including normal controls, were utilized (*n* = 498, TCGA). These data are acquired using NCBI’s GEO2R.

### 4.2. Mice

Mice, strain C57BL/6, male and female, were housed in microisolator cages under pathogen-free conditions at Tulane University School of Medicine. Both wildtype (WT) and FOXO3 knockout (FOXO3 KO) [87] mice had free access to standard chow diet and water. All littermates were genotyped to identify homozygous WT and FOXO3 KO [87], according to the guidelines of Tulane Institutional Animal Care and Use Committee.

Additionally, mice (57BL/6 strain) obtained from Jackson laboratory (6 weeks old) were housed at Tulane University School of Medicine. One group of mice were maintained on a standard chow diet and the other on a high-fat chow diet, 60% kcal/fat (D12492). For a period of 20 weeks, mice from both diet groups consistently gained weight; however, the weight of mice on a high-fat diet was more than 50% higher than mice on a regular diet [41]. Colons from these mice were utilized for protein extraction and transcriptomic analysis. Transcriptomes, previously acquired [41], from colons of high-fat-diet obese mice compared to regular diet (*n* = 3 for each group) are available through NCBI’s Sequence Read Archive (SRP093363).

Moreover, publicly available transcriptomes obtained from mice with colonic inflammation (*n* = 3) and dysplasia (*n* = 3) were utilized (GSE31106).

### 4.3. Mouse Peritoneal Macrophage

Experimental mice (six- to eight-weeks old) were injected intraperitoneally with 3% thioglycollate solution (Thermofisher, Waltham, MA, USA), which causes a peritoneal inflammatory response that allows for macrophage maturation [43,44]. After four days, mouse peritoneal cells, predominantly macrophages [43,88], were harvested from the abdominal cavity and pelleted (4 °C for 10 min).

### 4.4. RNA Isolation and cDNA Synthesis

Total RNA from harvested macrophages was isolated using the miRNeasy kit (Zymo Research, Irvibe, CA, USA), following the manufacturer’s instructions. RNA was assessed for quality by Agilent Bioanalyzer (Agilent Technologies, Santa Clara, CA, USA). Samples having RNA integrity numbers (RIN) more than 7 were utilized. RNA was then reverse transcribed to cDNA with qScipt cDNA SuperMix (Quantabio, Bevely, MA, USA).

### 4.5. qPCR

cDNA was generated from macrophages and used for qPCR as described previously [41]. The primers used for amplification of mouse cDNA are as follows: (mIL-1B-FOR 5′-TGCCACCTTTTGACAGTGATG-3′, mIL-1B-REV 5′-TTCTTGTGACCCTGAGCGAC-3′, mCXCR2-FOR 5′-GCTCACAAACAGCGTCGTAG-3′, mCXCR2-REV 5′-ATGGGCAGGGCCAGAATTAC-3′, mS100A8-FOR 5′-ATCCTTTGTCAGCTCCGTCTTC-3′, mS100A8-REV 5′-CTTCTCCAGTTCAGACGGCA-3′, mS100A9-FOR 5′-CTGCATGAGAACAACCCACG-3′, mS100A9-REV 5′-TCCCTTTAGACTTGGTTGGGC-3′, mTREM1-FOR 5′- ACAGAGGCAGTCGTTGGAG-3′, mTREM1-REV 5′-AGTGAACACATCTGAAGAACCTGAG-3′, mSTRA6-FOR 5′-GGTTCTTAAAGCAGGTGTGGG-3′, mSTRA6-REV 5′-ATGCTCCAGCTCTTCTTCCTAAC-3′, mVSIG10-FOR 5′-GGTTGAGTGTGAAAGAACCGC-3′, mVSIG10-REV 5′-GCGGTCTAAGTTCCCGTTGA-3′, mSERPINH1-FOR 5′-CCCGGCCCAGAATGAAAAAG-3′, mSERPINH1-REV 5′-TGGCTTTACCACCCAGTGAC-3′, mLAMB1-FOR 5′-GTGAGGAGAACAAAGTAGTTAAGCG-3′, mLAMB1-REV 5′-TGCCTGTCTTTTCTTCGGGT-3′, mNFE2L3-FOR 5′-TCTGTTGAGCTTGGTAGGGC-3′, mNFE2L3-REV 5′-CGAAGCCGAGAAGGGGTTAG-3′, mOLR1-FOR 5′-TGAAGCCTGCGAATGACGAG-3′, mOLR1-REV 5′-GGTTGGGAGACTTTGGAGGG-3′,mDNAJC28-FOR 5′-CCCATCACGTCTGTGAAGATCA-3′, mDNAJC28-REV 5′-GTTGGCGAAGAACTCCCTC-3′. The comparative Ct method was used to determine mRNA levels with GAPDH as a housekeeping control. cDNA was quantified using the C1000 Thermal Cycler system (Bio-Rad, Hercules, CA, USA) and PerfeCTa SYBR Green FastMix (Quantabio, USA).

### 4.6. Protein Extraction and Immunoblot

Protein extraction and immunoblots were performed as described previously [30,31,41]. The following specific antibodies against proteins were used: phosphorylated FOXO3 (Ser253) (Cell Signaling, Danvers, MA, USA) and β-actin (Cell Signaling). IRDye conjugated secondary antibodies (LI-COR, Lincoln, NE, USA) were used for protein visualization through the ChemiDoc MP imaging system (Bio-rad).

### 4.7. RNA Sequencing and Differential Expression Testing

RNA sequencing (RNA-seq) was performed as described before [38,41]. The sequencing data and the experiment design have been submitted to NCBI’s Sequence Read Archive, which are publicly available under study accession number GSE198058.

### 4.8. Transcriptome and Pathway Analysis

Data analysis for RNA-seq was performed using Ingenuity Pathway Analysis (IPA) (Qiagen, Germantown, MD). Differentially expressed genes (DEGs) meeting an expression threshold of >|1.5|-fold change relative to control and a false discovery rate (FDR) of less than 0.05 were entered into IPA. Clustered heatmaps of z-scaled transcripts per million (TPM) values for the top genes across all samples were generated using a Python data visualization package (Seaborn). TCGA network data was utilized for the expression of select transcripts (https://cistrome.shinyapps.io/timer, accessed on 10 October 2021) [89].

### 4.9. Hierarchical Clustering

Hierarchical clustering of transcriptomes from FOXO3-deficient macrophages relative to other experimental groups was performed using an uncentered correlation as a symmetric matrix, Pearson correlation for the similarity measure and complete linkage using Cluster3 software 38. The heatmaps were visualized with JavaTree software 39 (1.2.0 version, https://sourceforge.net/projects/jtreeview/files/jtreeview/1.2.0/ accessed on 10 October 2021)

### 4.10. Principal Component Analysis and Transcriptional Signature score

Principal component analysis (PCA) of FOXO3-deficient macrophage signatures with IBD (GSE4183) and human colon cancer (TCGA) transcriptomes was performed by FactoMineR R package with the PCA function. Percentage of variation and the first two coordinates of the above-mentioned samples were plotted. The following formula was utilized for the summary of multicohort signature in a single value:Score = mean[log2(x + 1 m)] where ‘x’ represents the expression of transcript, and ‘m’ represents the median of transcripts [90].

### 4.11. CIBERSORT

For quantification of macrophages in mouse colonic tissue, a computational method (CIBERSORT) was utilized for the assessment of RNA-seq [42]. For this purpose, mixture files of transcript per million from RNA-seq data were created, followed by generation of a specific murine immune cell and colonic cell signature along with phenotype files and reference in accordance to CIBERSORT specification (http://cibersort.stanford.edu, accessed on 10 May 2021) by employing the RNA-seq run accession numbers as previously described [38]. Values with a significance threshold (*p <* 0.05) were included in the analysis.

### 4.12. Statistical Analysis

All results are represented as means ± S.E. The statistical analysis of experiments was carried out by Student’s unpaired t-test or through ANOVA for one-war analysis of variance as well as Student–Newman–Keuls post-test in Graph Pad Software. A *p*-value of <0.05 was considered significant.

## Figures and Tables

**Figure 1 metabolites-12-00250-f001:**
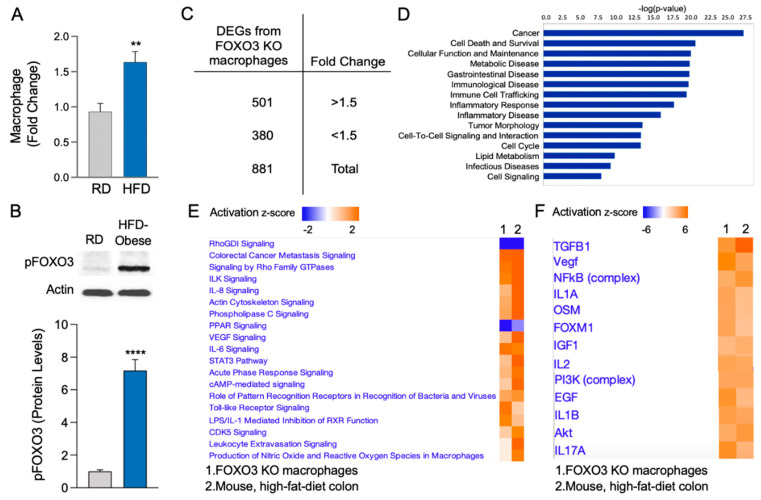
High-fat-diet obesity in mice results in an increased presence of colonic macrophages along with loss of FOXO3. (**A**) Transcriptomes from colon of high-fat-diet (HFD) obese mice showed augmented macrophage presence relative to control mice (regular diet, RD) colon (CIBERSORT, *n* = 3 per group, ** *p <* 0.01). (**B**) Colons of mice fed with HFD showed increased phosphorylation of FOXO3 compared to colons of control mice (RD). Actin was used as a loading control. Graph represents pFOXO3 densitometric quantification (*n* = 3 per group, **** *p <* 0.0001). (**C**) Differentially expressed genes (DEGs) from FOXO3-deficient macrophages relative to control (*n* = 3 for each group, FC > |1.5|, FDR < 0.05). (**D**) Top diseases and pathways affected by FOXO3-deficient macrophages relative to control (*p <* 0.05, IPA). (**E**,**F**) Top canonical pathways and upstream regulators affected by FOXO3-deficient macrophages, which are altered in HFD obese mouse colon (SRP093363) (*n* = 3 per group, *p <* 0.05, IPA).

**Figure 2 metabolites-12-00250-f002:**
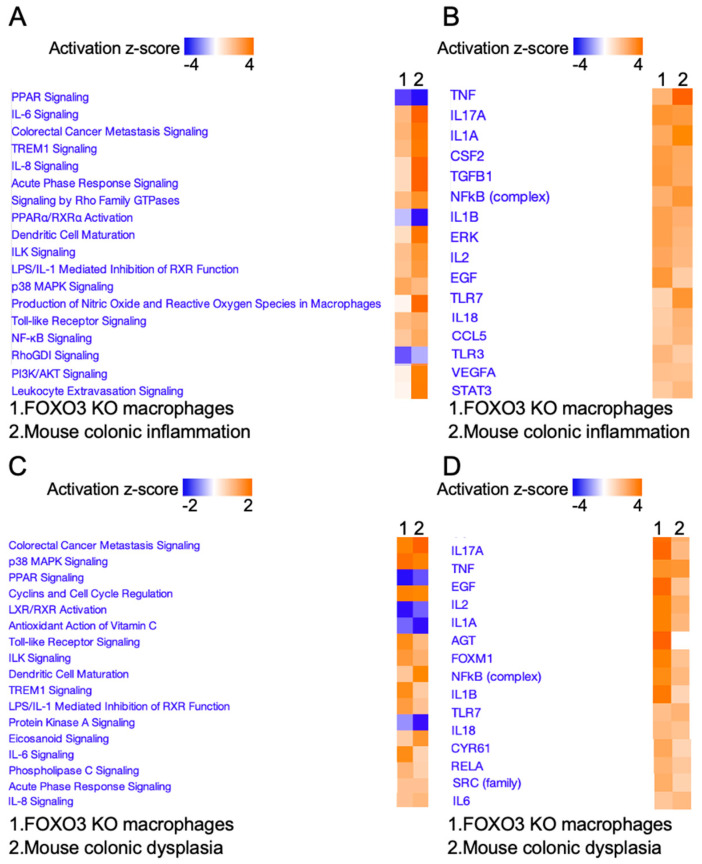
FOXO3 deficiency in macrophages links to mouse colonic inflammation and dysplasia. (**A**,**B**) Top canonical pathways and upstream regulators affected by FOXO3-deficient macrophages which are altered in mouse inflamed colonic tissue (*n* = 3, GSE31106, *p <* 0.05, IPA). (**C**,**D**) Top canonical pathways and upstream regulators affected by FOXO3-deficient macrophages which are altered in mouse dysplastic colonic tissue (*n* = 3, GSE31106, *p <* 0.05, IPA).

**Figure 3 metabolites-12-00250-f003:**
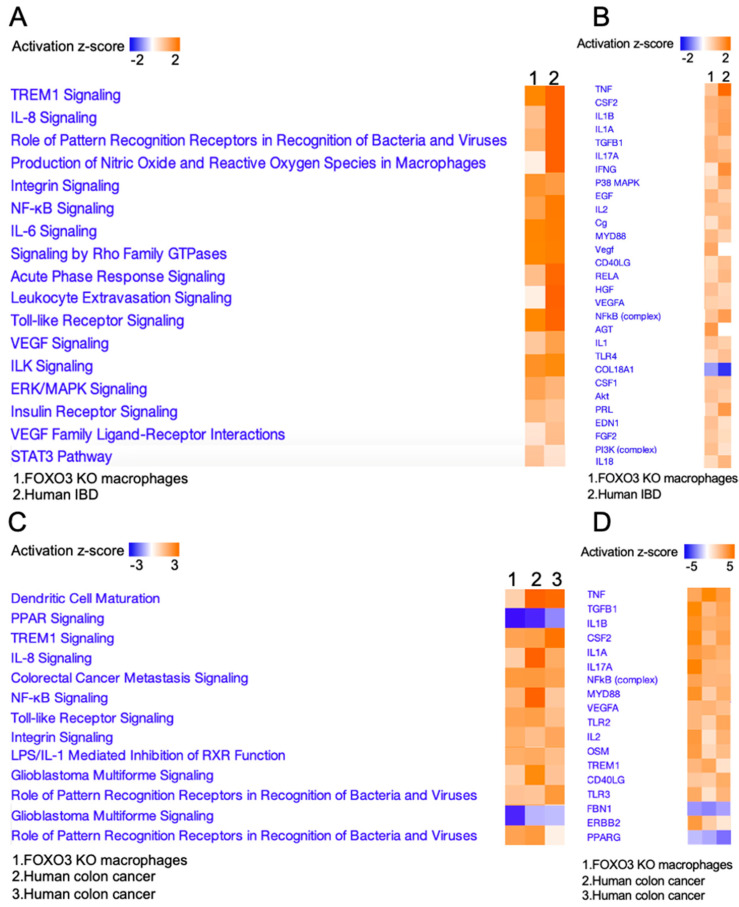
Transcriptome of FOXO3-deficient macrophages is significantly prevalent in IBD and human colon cancer. (**A**,**B**) Similar pathways and upstream regulators associated with DEGs representing FOXO3 deficient macrophages and IBD (*n* = 23, GSE4183, *p <* 0.05, IPA). (**C**,**D**) Similar pathways and upstream regulators associated with DEGs representing FOXO3-deficient macrophages and human colon cancer relative to normal colon (*n* = 29, GSE4183, GSE141174, *p <* 0.05, IPA).

**Figure 4 metabolites-12-00250-f004:**
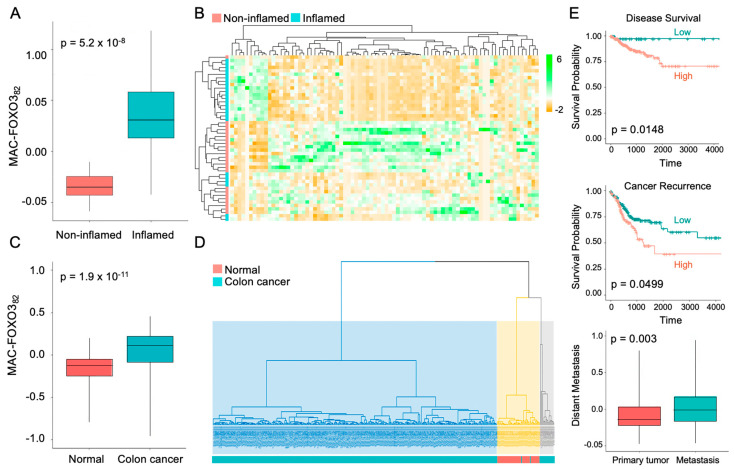
MAC-FOXO3_82_ signature in IBD and colon cancer. (**A**,**B**) Principal component analysis (PCA) of inflamed IBD and matched non-inflamed control transcriptomes with the MAC-FOXO3_82_ signature was performed to estimate variation between samples. Two-axis values of the PCA showed the MAC-FOXO3_82_ significantly differentiated inflamed IBD from matched non-inflamed control tissue. DEGs representing MAC-FOXO3_82_ signature on y-axis and IBD samples on x-axis (*n* = 23, GSE4183, *p* = 5.2 × 10^−8^). Hierarchical clustering, as shown by representative heatmap, revealed two distinct clusters of IBD samples separated by MAC-FOXO3_82,_ differentiating between inflamed IBD and matched non-inflamed control transcriptomes. DEGs representing MAC-FOXO3_82_ signature on x-axis and human IBD samples on y-axis (*n* = 23, GSE4183). (**C**,**D**) Principal component analysis (PCA) of human colon cancer and matched normal (control) colonic tissue transcriptomes with the MAC-FOXO3_82_ signature was performed to estimate variation between samples. Two-axis values of the PCA showed MAC-FOXO3_82_ significantly differentiated human colon cancer from matched normal (control tissue) (*n* = 498, TCGA, *p* = 1.9 × 10^−11^). Hierarchical clustering, as shown by representative heatmap, revealed two distinct clusters of human colon cancer samples separated by MAC-FOXO3_82_ differentiating between human colon cancer and matched normal control transcriptomes (*n* = 498, TCGA). (**E**) Increased MAC-FOXO3_82_ signature presence in colon cancer patients is associated with poor survival rates (*p =* 0.0148), increased risk of cancer recurrence (*p =* 0.0499), and distant metastasis (*p =* 0.003) (Kaplan–Meier survival analysis).

**Figure 5 metabolites-12-00250-f005:**
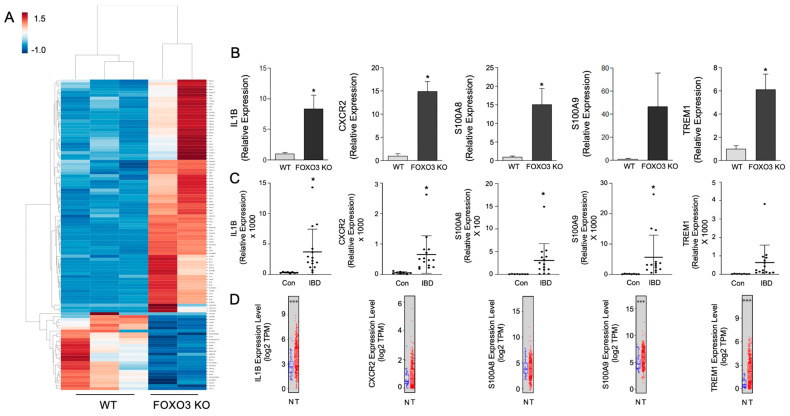
FOXO3-mediated differentially expressed genes in macrophages with established roles in IBD and colon cancer. (**A**) A heatmap of the top DEGs specific to FOXO3-deficient peritoneal macrophages relative to control (>|1.5|-fold change, FDR < 0.05). (**B**) Validation of select FOXO3 dependent IL-1B, CXCR2, S100A8, S100A9 and TREM1 transcripts in macrophages (qPCR, *n* = 3, * *p <* 0.05). (**C**,**D**) Altered IL-1B, CXCR2, S100A8, S100A9 and TREM1 levels in IBD and human colon cancer patient transcriptomes (*n* = 23, GSE4183; *n* = 498, TCGA, * *p <* 0.05, *** *p <* 0.001).

**Figure 6 metabolites-12-00250-f006:**
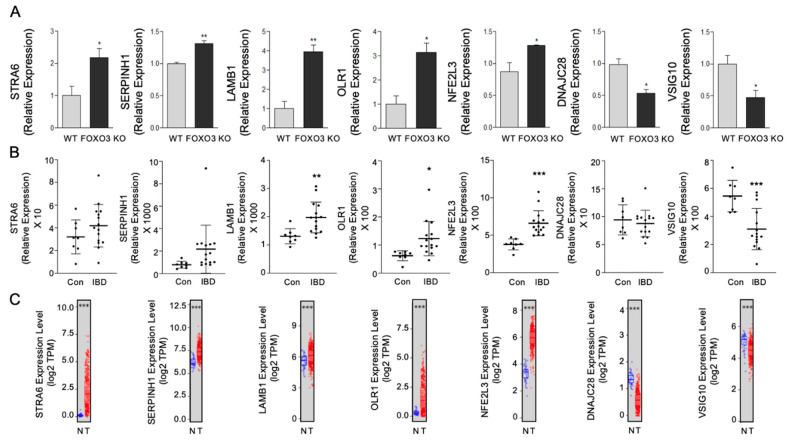
FOXO3-mediated differentially expressed genes in macrophages with unexplored roles in IBD and colon cancer. (**A**) Validation of select FOXO3-dependent STRA6, SERPINH1, LAMB1, OLR1, NFE2L3, DNAJC28 and VSIG10 transcripts in macrophages (qPCR, *n* = 3, * *p <* 0.05, ** *p <* 0.01). (**B**,**C**) Altered STRA6, SERPINH1, LAMB1, OLR1, NFE2L3, DNAJC28 and VSIG10 levels in human IBD and colon cancer patient transcriptomes (*n* = 23, GSE4183; *n* = 498, TCGA, * *p <* 0.05, ** *p <* 0.01, *** *p <* 0.001).

**Table 1 metabolites-12-00250-t001:** Differentially expressed genes of MAC-FOXO3_82_ in FOXO3-deficient mouse macrophages relative to control (*n* = 3, FC ≥|1.5|, *p <* 0.001).

	Gene	Gene Name	FC	*p*-Value
1	*Stfa1*	Stefin-A 1	225.6	4.8 × 10^−^^8^
2	*Entpd3*	Ectonucleoside Triphosphate Diphosphohydrolase 3	113.6	3.0 × 10^−^^9^
3	*Amer2*	APC Membrane Recruitment Protein 2	67.7	7.6 × 10^−^^10^
4	*S100a9*	S100 Calcium Binding Protein A9	66.7	1.1 × 10^−^^4^
5	*Mrgpra2b*	MAS Related GPR Family Member X2	35.3	2.4 × 10^−^^4^
6	*Asprv1*	Aspartic Peptidase Retroviral-like 1	33.9	7.5 × 10^−^^18^
7	*Sycp2*	Synaptonemal Complex Protein 2	30.2	1.6 × 10^−^^13^
8	*Chil1*	Chitinase 3-like 1	29.5	1.4 × 10^−^^6^
9	*Olfm4*	Olfactomedin 4	24.3	6.2 × 10^−^^8^
10	*Cxcr2*	C-X-C Motif Chemokine Receptor 2	21.5	8.4 × 10^−^^35^
11	*Catspere2*	Catsper Channel Auxiliary Subunit Epsilon	18.6	9.7 × 10^−^^5^
12	*Il1f9*	Interleukin 1 Family, Member 9	13.6	5.9 × 10^−^^13^
13	*Amd2*	Adenosylmethionine Decarboxylase 1 Pseudogene 2	11.8	2.6 × 10^−^^34^
14	*Alas2*	5′-Aminolevulinate Synthase 2	10.1	1.9 × 10^−^^5^
15	*S100a8*	S100 Calcium-binding Protein A8	9.9	4.9 × 10^−^^7^
16	*Steap4*	Six-Transmembrane Epithelial Antigen Of Prostate 4	8.0	2.1 × 10^−^^12^
17	*Il1r2*	Interleukin 1 Receptor Type 2	7.9	6.6 × 10^−^^8^
18	*Hba-a2*	Hemoglobin Subunit Alpha 2	7.7	3.1 × 10^−^^6^
19	*Slc38a4*	Solute Carrier Family 38 Member 4	7.7	3.9 × 10^−^^8^
20	*Stra6*	Stimulated By Retinoic Acid 6	7.2	3.7 × 10^−^^5^
21	*Col24a1*	Collagen Type XXIV Alpha 1 Chain	6.7	9.3 × 10^−^^6^
22	*Hba-a1*	Hemoglobin Subunit Alpha 1	5.9	1.7 × 10^−^^7^
23	*Trem1*	Triggering Receptor Expressed On Myeloid Cells 1	5.7	1.2 × 10^−^^15^
24	*Lin28a*	Lin-28 Homolog A	5.6	1.5 × 10^−4^
25	*Il1b*	Interleukin 1 Beta	5.4	1.5 × 10^−^^9^
26	*Ambp*	Alpha-1-Microglobulin/Bikunin Precursor	5.1	2.3 × 10^−^^4^
27	*Ifitm1*	Interferon-induced Transmembrane Protein 1	5.1	5.8 × 10^−^^5^
28	*Nfe2l3*	Nuclear Factor, Erythroid 2-like 3	4.9	1.9 × 10^−^^5^
29	*Kirrel*	Kin Of Irregular Chiasm-like Protein 1	4.9	9.2 × 10^−^^5^
30	*Lamb1*	Laminin Subunit Beta 1	4.3	6.0 × 10^−^^7^
31	*Serpinh1*	Serpin Peptidase Inhibitor, Clade H, Member 1,	4.0	1.7 × 10^−^^6^
32	*Adamts1*	ADAM Metallopeptidase With Thrombospondin Type 1 Motif 1	3.9	1.8 × 10^−^^4^
33	*Pkhd1l1*	Polycystic Kidney and Hepatic Disease 1-like 1	3.8	1.8 × 10^−^^4^
34	*Fads2*	Fatty Acid Desaturase 2	3.7	6.2 × 10^−^^6^
35	*Col1a1*	Collagen Type I Alpha 1 Chain	3.6	2.5 × 10^−^^4^
36	*Dmkn*	Dermokine	3.6	4.0 × 10^−^^5^
37	*Rbp1*	Retinol-binding Protein 1	3.5	7.2 × 10^−^^5^
38	*Tcaf1*	TRPM8 Channel Associated Factor 1	3.4	9.0 × 10^−^^8^
39	*Hpgd*	15-Hydroxyprostaglandin Dehydrogenase	3.4	2.1 × 10^−^^7^
40	*septin3*	Neuronal-specific Septin-3	3.4	6.2 × 10^−^^5^
41	*Nr4a3*	Nuclear Receptor Subfamily 4 Group A Member 3	3.3	5.8 × 10^−^^7^
42	*Ccl2*	C-C Motif Chemokine Ligand 2	3.3	3.1 × 10^−^^5^
43	*Ltbp2*	Latent Transforming Growth Factor Beta-binding Protein 2	3.3	1.6 × 10^−^^4^
44	*Dclk1*	Doublecortin-like Kinase 1	3.2	3.6 × 10^−^^6^
45	*Map1b*	Microtubule Associated Protein 1B	3.2	2.2 × 10^−^^4^
46	*Wt1*	Wilms Tumor 1	3.2	4.1 × 10^−^^5^
47	*Col1a2*	Collagen Type I Alpha 2 Chain	3.1	1.9 × 10^−^^4^
48	*Ptprf*	Protein Tyrosine Phosphatase Receptor Type F	3.0	2.2 × 10^−^^4^
49	*Krt19*	Keratin 19	2.9	1.0 × 10^−^^4^
50	*Arhgef17*	Rho-specific Guanine-Nucleotide Exchange Factor	2.8	1.5 × 10^−^^4^
51	*Il1r1*	Interleukin 1 Receptor Type 1	2.7	6.3 × 10^−^^5^
52	*Treml4*	Triggering Receptor Expressed On Myeloid Cells-like 4	2.7	1.9 × 10^−^^4^
53	*Vcan*	Versican	2.5	1.3 × 10^−^^5^
54	*Klrb1b*	Killer Cell Lectin-like Receptor B1	2.4	2.9 × 10^−^^5^
55	*Olr1*	Oxidized Low Density Lipoprotein Receptor 1	2.1	2.8 × 10^−^^4^
56	*Pram1*	PML-RARA Regulated Adaptor Molecule 1	2.1	2.2 × 10^−^^5^
57	*Ccnb1*	Cyclin B1	2.1	1.9 × 10^−^^4^
58	*Npl*	N-Acetylneuraminate Pyruvate Lyase	2.0	6.2 × 10^−^^5^
59	*Cdk1*	Cyclin-dependent Kinase 1	1.9	1.5 × 10^−^^4^
60	*Tbc1d16*	TBC1 Domain Family Member 16	1.8	7.4 × 10^−^^5^
61	*Dab2*	DAB Adaptor Protein 2	1.8	3.1 × 10^−^^7^
62	*Nfil3*	Nuclear Factor, Interleukin 3 Regulated	1.8	2.9 × 10^−^^5^
63	*Tbc1d4*	TBC1 Domain Family Member 4	1.7	1.2 × 10^−^^4^
64	*Samsn1*	SAM Domain, SH3 Domain And Nuclear Localization Signals 1	1.5	2.6 × 10^−^^5^
65	*Kcnk13*	Potassium Two Pore Domain Channel Subfamily K Member 13	1.5	1.4 × 10^−^^4^
66	*Fmo5*	Flavin Containing Dimethylaniline Monoxygenase 5	−1.6	1.2 × 10^−^^4^
67	*Ip6k1*	Inositol Hexakisphosphate Kinase 1	−1.6	8.4 × 10^−^^5^
68	*Wdr45*	WD Repeat Domain 45	−1.7	2.2 × 10^−^^5^
69	*Serinc5*	Serine Incorporator 5	−1.7	3.2 × 10^−^^5^
70	*Mrgprx2*	MAS-Related GPR Family Member X2	−1.9	3.7 × 10^−^^1^
71	*Maml2*	Mastermind-like Transcriptional Coactivator 2	−1.9	7.2 × 10^−^^5^
72	*Vsig10*	V-Set And Immunoglobulin Domain-containing 10	−2.2	4.5 × 10^−^^5^
73	*Dnajc28*	DnaJ Heat Shock Protein Family (Hsp40) Member C28	−2.2	7.9 × 10^−^^7^
74	*Rab6b*	RAB6B, Member RAS Oncogene Family	−2.5	2.5 × 10^−^^5^
75	*Ypel3*	Yippee Like 3	−2.9	4.2 × 10^−^^19^
76	*Armc2*	Armadillo Repeat-containing 2	−3.1	2.8 × 10^−^^6^
77	*Cdkn2a*	Cyclin-dependent Kinase Inhibitor 2A	−5.5	8.4 × 10^−^^6^
78	*Slc15a2*	Solute Carrier Family 15 Member 2	−6.6	1.0 × 10^−^^4^
79	*Clec2g*	C-Type Lectin Domain Family 2 Member D	−8.9	2.1 × 10^−^^20^
80	*Camk2b*	Calcium/Calmodulin-dependent Protein Kinase II Beta	−9.4	3.2 × 10^−^^11^
81	*Slc25a27*	Solute Carrier Family 25 Member 27	−9.4	4.3 × 10^−^^6^
82	*Foxo3a*	Forkhead Box O3	−22.0	8.1 × 10^−^^43^

## Data Availability

The data presented in this study are available on request from the corresponding author. The data are not publicly available due to privacy restrictions.

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
