# Peer review of "FOXO3 Expression in Macrophages Is Lowered by a High-Fat Diet and Regulates Colonic Inflammation and Tumorigenesis"

_metabolites, 2022, doi:10.3390/metabo12030250_

Round 1
Reviewer 1 Report
This manuscript if of interest but the title for this manuscript does not accurately describe the content. From the way the manuscript is written it is unclear if the transcriptomic data were obtained from previously published work. Moreover, additional information should be included or at least cited to direct the reader to the previously published work: Dietary content, age of mice when the diet started, duration of diet, change in body weight of mice over duration of diet.
Much of the manuscript over concludes associations from publicly available databases as compared to macrophages from mice. (i.e.; "These data demonstrated that FOXO3 deficiency in macrophages may contribute contributes to colonic inflammatory and tumorigenic pathobiologies." and "Together, these data demonstrated significance of macrophages-FOXO3 axis both in human IBD and colon cancer.") These data demonstrate associations of transcriptomic signatures at best.
Author Response
1. This manuscript if of interest but the title for this manuscript does not accurately describe the content.
-We agree with this comment. Thus, the new title is provided “Loss of FOXO3 in macrophages contributes to high-fat-diet obesity-mediated colonic inflammation and tumorigenesis with a possible link to inflammatory bowel disease and colon cancer”. (Page 1).
2. From the way the manuscript is written it is unclear if the transcriptomic data were obtained from previously published work.
-Thank you for bringing this to our attention. We clarified in the manuscript that transcriptomic data from HFD-obese mouse colon were previously obtained and have provided the SRP093363 number and references publication in Carcinogenesis 2017. Further, the body weight of mice at the end of the experiment has been included: “First, we utilized transcriptomic data obtained from the colon of both HFD-obese mice and mice fed with regular diet (RD) (SRP093363). As we described before [41], consistent with an obese phenotype, mice on HFD gained on average 54±2* g while mice on RD gained 33±3 g during a period of 20 weeks (*p<0.05). From these mice, we utilized transcriptomes obtained from their colon [41] to assess macrophage levels with CIBERSORT [42]. This is a platform that employs the transcriptional signatures of specific immune cells to determine their levels in the tissue. There was an abundance of macrophages in the colon of HFD-obese mice relative to control (RD) (Figure 1A). Moreover, we assessed scraped mucosa from the colons of HFD-obese and RD-control mice for FOXO3 status. We found increased phosphorylated (inactive) FOXO3 levels in HFD-obese mice colonic mucosa relative to control (Figure 1B). Therefore, as HFD-obesity cause increased macrophages as well as loss of FOXO3 in mouse colon, we determined that the loss of FOXO3 in macrophages contributes to obesity-mediated colonic pathobiology. We obtained intraperitoneal macrophages [43,44] from FOXO3 knockout (KO) and wildtype (WT) mice in order to determine systemic transcriptional changes (RNA-seq). The population of these induced peritoneal macrophages included a slight number of other immune cells [43,45] which, after further sequencing and bioinformatic analysis, proved to be insignificant. Transcriptional assessment showed significant number of differentially expressed genes (DEGs) in macrophages obtained from FOXO3 deficient mice relative to control (WT). Among these DEGs, 501 were increased and 380 were decreased (Figure 1C) (>|1.5|-fold change, FDR<0.05).” (Page 2-3).
3. Moreover, additional information should be included or at least cited to direct the reader to the previously published work: Dietary content, age of mice when the diet started, duration of diet, change in body weight of mice over duration of diet.
-This information was provided in the Methods: “Additionally, mice (57BL/6 strain) obtained from Jackson laboratory (6 weeks old), were housed at Tulane University School of Medicine. One group of mice were maintained on a standard chow diet and the other on a high-fat chow diet, 60% kcal/fat (D12492). For period of 20 weeks, mice from both diet groups consistently gained weight; however, the weight of mice on high-fat diet was more then 50% higher than mice on regular diet [41]. Colons from these mice were utilized for protein extraction and transcriptomic analysis. Transcriptomes, previously acquired [41], from colon of high-fat-diet obese mice compared to regular diet (n=3 for each group) are available through NCBI’s Sequence Read Archive (SRP093363).” (Page 13).
4. Much of the manuscript over concludes associations from publicly available databases as compared to macrophages from mice. (i.e.; "These data demonstrated that FOXO3 deficiency in macrophages may contribute contributes to colonic inflammatory and tumorigenic pathobiologies." and "Together, these data demonstrated significance of macrophages-FOXO3 axis both in human IBD and colon cancer.") These data demonstrate associations of transcriptomic signatures at best.
- The overstated findings have been revised in the manuscript.
”These data demonstrated that FOXO3 deficiency in macrophages is associated with colonic inflammatory and tumorigenic pathobiologies.” (Page 4).
” Together, these data demonstrated that the macrophage-FOXO3 axis is associated with both human IBD and colon cancer.” (Page 6).
Reviewer 2 Report
The authors presented every comprehensive study, where they have found that in mouse colon, high-fat-diet-(HFD)-obesity led to diminished FOXO3 levels and increased macrophages. Additionally, authors have identified a specific transcriptional signature for the macrophage-FOXO3 axis (MAC- FOXO382), which separated the transcriptome of affected tissue from control in both IBD (p=5.2E- 08) and colon cancer (p=1.9E-11) revealing its significance in human colonic pathobiologies. Outcomes of this study lead next step to understanding of how HFD-obesity and related metabolites promote colonic pathobiologies.
IMHO the manuscript should be published after minor revisions.
My remark to manuscript:
- summary/conclusions paragraph of this stud is lacking - it should be added at the end of paper and summarize key findings and underline the further perspectives/further studies
Author Response
1. Summary/conclusions paragraph of this stud is lacking - it should be added at the end of paper and summarize key findings and underline the further perspectives/further studies
- The statements of the findings and future perspectives have been included in the last parapgraph of the Discussion: “Here we showed that obesity, whether directly or indirectly, drives colonic inflammation and tumorigenesis through the inactivation of FOXO3 in macrophages. Additionally, we established the significance of the macrophage-FOXO3 axis, in both IBD and colon cancer. In future studies, the novel differentially expressed genes that we have identified, which are FOXO3-dependent in macrophages, will be examined for their roles in these colonic pathobiologies.” (Page 13).
Round 2
Reviewer 1 Report
The authors have addressed my concerns.
Author Response
Thank you for taking the time to read our manuscript. We appreciate your thoughtful suggestions. The revised manuscript was edited several times to improve the style and grammar/spelling.